# Short-Term Effects of Side-Stream Smoke on Nerve Growth Factor and Its Receptors TrKA and p75^NTR^ in a Group of Non-Smokers

**DOI:** 10.3390/ijerph191610317

**Published:** 2022-08-19

**Authors:** Anna Maria Stabile, Alessandra Pistilli, Desirée Bartolini, Eleonora Angelucci, Marco Dell’Omo, Gabriele Di Sante, Mario Rende

**Affiliations:** 1Department of Medicine and Surgery, Section of Human Anatomy, Clinical and Forensic, School of Medicine, University of Perugia, P. le Lucio Severi 1 Sant’Andrea delle Fratte, 06132 Perugia, Italy; 2Department of Pharmaceutical Sciences, Section of Biochemistry, University of Perugia, Via del Giochetto, 06132 Perugia, Italy; 3Department of Medicine, Section of Occupational Medicine, Respiratory Diseases and Toxicology, School of Medicine, University of Perugia, Lucio Severi 1 Sant’Andrea delle Fratte, 06132 Perugia, Italy

**Keywords:** side-stream smoke, nerve growth factor, neurotrophin receptor p75, tropomyosin-related kinase A, cotinine, second-hand smoke, environmental tobacco smoke

## Abstract

Environmental tobacco smoke remains a major risk factor, for both smokers and non-smokers, able to trigger the initiation and/or the progression of several human diseases. Although in recent times governments have acted with the aim of banning or strongly reducing its impact within public places and common spaces, environmental tobacco smoke remains a major pollutant in private places, such as the home environment or cars. Several inflammatory and long-term biomarkers have been analysed and well-described, but the list of mediators modulated during the early phases of inhalation of environmental tobacco smoke needs to be expanded. The aim of this study was to measure the short-term effects after exposure to side-stream smoke on Nerve Growth Factor and its receptors Tropomyosin-related kinase A and neurotrophin p75, molecules already described in health conditions and respiratory diseases. Twenty-one non-smokers were exposed to a home-standardized level of SS as well as to control smoke-free air. Nerve Growth Factor and inflammatory cytokines levels, as well the expression of Tropomyosin-related kinase A and neurotrophin receptor p75, were analysed in white blood cells. The present study demonstrates that during early phases, side-stream smoke exposure induced increases in the percentage of neurotrophin receptor p75-positive white blood cells, in their mean fluorescent intensity, and in gene expression. In addition, we found a positive correlation between the urine cotinine level and the percentage of neurotrophin receptor-positive white blood cells. For the first time, the evidence that short-term exposure to side-stream smoke is able to increase neurotrophin receptor p75 expression confirms the very early involvement of this receptor, not only among active smokers but also among non-smokers exposed to SS. Furthermore, the correlation between cotinine levels in urine and the increase in neurotrophin receptor p75-positive white blood cells could represent a potential novel molecule to be investigated for the detection of SS exposure at early time points.

## 1. Introduction

Environmental tobacco smoke (ETS) is composed of complex and reactive chemical compounds that mix, interact, and can be diluted into environmental air. ETS consists of two components: the mainstream smoke exhaled by smokers and narrowly defined Second-hand Smoke (SHS) and Side-stream Smoke (SS), emitted from the direct burning of tobacco, which constitutes approximately 85% of ETS [1,2,3,4]. Although conventionally and synecdochally SHS is used to indicate both components of ETS, in this experimental study, we will refer only to SS.

Growing evidence about the relevant health risk among non-smokers exposed to ETS led several countries to implement stringent smoke-free regulations in workplaces and public areas. Conversely, ETS exposure in private environments, particularly at home or in cars, cannot be regulated by any smoke-free legislation. Therefore, an unmet need for public health is to study the impact of ETS exposure in domestic ambientes on healthcare, mortality, and social welfare spending [5,6,7,8,9]. Several clinical studies have shown that ETS represents a risk factor for lung cancer [10,11] and induces the release of several cytokines involved in phlogistic processes [12,13], such as inflammatory airway diseases. In line with these findings, we have previously demonstrated an interesting role of Nerve Growth Factor (NGF) and its receptors Tropomyosin-related kinase A (TrKA) and neurotrophin receptor p75 (p75^NTR^) in the most typical smoke-induced disease: Chronic Obstructive Pulmonary Disease (COPD) [14]. Furthermore, in a human bronchial epithelial cell line (BEAS), we have shown that chronic exposure to cigarette smoke extract is able to induce the release of NGF and to increase the expression of p75^NTR^ [15].

NGF, a member of the neurotrophin family, interacts mainly with two distinct cell surface receptors: the specific high-affinity TrKA and the common low-affinity p75^NTR^ [16,17]. Besides its classical role in neuronal growth and survival, NGF has shown pleiotropic effects on various populations of non-neuronal cells, including cancer and immune cells [18,19,20,21]. B and T lymphocytes, as well as mast cells, monocytes, dendritic cells, and macrophages, although with cell-dependent variability, are capable of releasing high amounts of NGF and expressing both TrKA and p75^NTR^ receptors [14,22,23,24,25].

Although the role of NGF and its receptors have been described in several diseases and conditions, such as tumour progression [21,26], inflammatory and immune processes [25], and chronic smoke exposure [15], the mechanisms underlying these correlations need to be expanded. In this context, the aim of this study is to determine the short-term effects of a one-hour exposure to SS on NGF and its receptors in peripheral blood of healthy non-smokers located in an environmentally controlled home-like room, focusing on the relationship among NGF, its receptors, and inflammatory cytokine expression levels.

## 2. Materials and Methods

If not otherwise specified, all chemicals were purchased from Sigma Aldrich (St. Louis, MI, USA).

### 2.1. Study Participants

The study comprised twenty-one healthy subjects. The excluding criteria were: smoking habit, exposure to SS in their own home environment, atopy (with a diagnosis of allergy with respiratory symptoms), treatment with any type of medication in the last three months, and diagnosis of any respiratory tract disease/symptoms for less than 8 weeks. Their body mass index is reported in Table 1. Before the study entry, all subjects observed at least a 12 h abstinence from meals, alcohol, and any type of beverages with coffee. Their informed consent was obtained in accordance with the Declaration of Helsinki and as part of the protocol approved by our institutional ethical committee (University of Perugia, NR.: 2016-08).

### 2.2. Home-like Room Characteristics and SS Protocol

The exposure of the subjects to SS was performed in a 6 × 5 × 4 m (120 m^3^) room (named the SS room), with environmentally controlled and stable parameters, such as air temperature (24 °C), air velocity (0.05 ms^−1^), and humidity (45%). An adjacent room with similar dimensions and environmental parameters was used as the Smoke-Free Air control room (named the SFA room). A restroom adjacent to both rooms was dedicated for urine collection by participants.

SS was generated by burning commercial cigarettes (Philip Morris, Marlboro Gold Original King-Size Hard Pack 20’s, with a declared content for each cigarette of 0.6 mg of nicotine and 8.0 mg of tar).

A total of twenty-four cigarettes were used in each experimental set. A first set of twelve cigarettes, distributed in two ashtrays located on a table at the centre of the SS room, were lit and allowed to burn until they reached the filter. Successively, a second set of twelve cigarettes replaced the first one to ensure and maintain a constant concentration of carbon monoxide (CO). Two ceiling fans were used to provide regular air circulation throughout the room. The CO concentration was determined using a CO-CO_2_ analyser (AQ 5000 Pro, Quest Technologies). The basal/background CO level within the SS room was less than 0.8 ppm. SS was adjusted to reach and maintain in the room a constant CO concentration of 14 ppm, verified via continuous measurements [27].

All the participants (Table 1) were initially gathered into the SFA room, where they were invited to bring their first morning urine void (control urine) collected in a polyethylene sample jar (Fisher Scientific, Thermo Fisher Scientific, Inc., Waltham, MA, USA). Afterwards, a certified phlebotomist applied a venous catheter in their antecubital vein to facilitate subsequent blood samplings. In this control room, the basal blood sample was collected (time point 0, T0). After a period of resting in the SFA room (about 20 min), four control subjects remained in the SFA room with a phlebotomist, while seventeen subjects entered the SS room, along with their phlebotomist, as soon as the desired above-described CO concentration of 14 ppm was obtained from the burning cigarettes. In both rooms, blood samples were drawn after 0.5 and 1 h (T1 and T2). After 1 h, all the participants were initially invited to use the restroom to collect a sample of their urine, and the SS-exposed subjects then joined the control subjects in the SFA room, where they remained for another 1 h. At the end of this period, the fourth blood samples were drawn (2 h, T3), the venous catheter was removed, and a further sample of urine was collected. All participants returned the following day for the last blood sampling and to consign their first morning urine void (24 h, T4).

### 2.3. Peripheral Blood Collection and White Blood Cells Isolation

At each time point investigated (0, 0.5, 1, 2 and 24 h), venous blood samples were collected in vacuum tubes (Vacutainer) containing EDTA, and white blood cells (WBCs) were isolated by centrifugation (400× *g* for 10 min). For the isolation of WBCs, blood samples were incubated for 10 min with a lysis buffer (155 mM NH_4_Cl, 10 mM KHCO_3_ and 0.1 mM EDTA, pH 7.4). Then, after washing, cells were resuspended in staining buffer (PBS plus BSA 0.5%, PBS/BSA).

### 2.4. Serum Collection

At each time point investigated (0, 0.5, 1, 2 and 24 h), venous blood samples were collected in vacuum tubes (Vacutainer) containing separator gel. Blood samples were centrifuged at 400× *g* for 10 min, and the cell-free serum supernatant was frozen at −80 °C.

### 2.5. Liquid Chromatography–Mass Spectrometry (LC-MS) Analysis of Urine Cotinine

At 0, 1, 2 and 24 h, urine samples were collected and immediately frozen at −20 °C. The LC-MS analysis was carried out using a Waters 2795 system coupled to a Quattro Premier XE MS tandem detector, equipped with an electron spray ionization (ESI) source. The column used was a Waters Atlantis dC18 (2.0 × 100 mm, 3 μm) with an inline guard column filter; the flow rate was set to 0.2 mL/min; the column temperature was set at 35 °C. The analyte was eluted using a mobile phase consisting of 50 mM (pH 5) ammonium formiate (mobile phase A) and acetonitrile (mobile phase B). The standard curve was prepared by dissolving cotinine in the mobile phase. Then, 10 μL of samples and standards pre-diluted 1:5 with mobile phase A were injected. The positive ESI MS/MS signal of cotinine was acquired as follows: capillary voltage at 1.0 kV, cone voltage at 35 V, source temperature at 130 °C, and desolvation temperature at 400 °C. The cone and desolvation gas flows were set at 50 and 500 L/h, respectively. The collision gas was argon at a collision cell pressure of 4 × 10^−3^ mbar. The quantification transition was 177.0 to 97.9 m7z, and the collision energy 20 eV. We used the areas under the peak for quantification [15]. The limit of detection (LOD) for cotinine was 0.0186 ng/mL. The cut-off sensitivity limit for cotinine amounted to 0.0620 ng/mL. Urinary cotinine concentrations were expressed as nanograms per gram creatinine in order to normalize the result for the different dilution grades of the urine samples [28].

### 2.6. Serum NGF Quantification

The amount of NGF in each serum sample was determined using an ELISA kit (D1000B, R&D System, Minneapolis, MN, USA) as previously described in Stabile et al. [20].

### 2.7. TrKA and p75^NTR^ Analysis by FACS

For FACS analysis, 1 × 10^6^ cells/mL freshly purified WBCs in PBS/BSA were stained with the following fluorochrome-labelled mAbs: FITC-labelled rabbit anti-human p75^NTR^ extracellular domain (ANT-007-F, Alomone Labs, Jerusalem, Israel) (10 mL/sample) and PE-labelled mouse anti-human TrKA (FAB1751P, R&D System, Minneapolis, MN, USA) (10 mL/sample). After 1 h, cells were washed, resuspended in staining buffer (PBS + 2% FBS + 1% paraformaldehyde), and analysed by FACS. All procedures were performed at 4 °C. A FACS Calibur (Becton Dickinson, Franklin Lakes, NJ, USA) equipped with a single 15 mW argon ion laser operating at 488 and 633 nm and interfaced with CellQuest Software (Becton Dickinson, Franklin Lakes, NJ, USA) was used. Samples were stained, compensated, and analysed with isotype controls. The data obtained from flow cytometer software were expressed as frequency of positive cells/antibody (% positive cells) and as mean fluorescent intensity (MFI) [25]. Gating strategy for the identification of cell subsets is described in Appendix A.

### 2.8. TrKA and p75^NTR^ and Cytokines Gene Expression RT-PCR

Total RNA was extracted following the recommendations of the miRNeasy Micro kit (no. 217084 Qiagen, Hilden, Germany). Using the enzyme SuperScript III (no. 18080-051 Invitrogen, Thermo Fisher Scientific, Inc., Waltham, MA, USA), 1 μg of total RNA was retrotranscribed into first-strand cDNA. qPCR was conducted on a StepOne System (Applied Biosystems, Thermo Fisher Scientific, Inc., Waltham, MA, USA) using a PowerUp SYBR Green Real-time PCR Master Mix (A25742 Applied Biosystems, Thermo Fisher Scientific, Inc., Waltham, MA, USA). The primers are shown in Appendix A and, for the relative quantification of gene expression, were normalized to GAPDH and actine housekeeping genes [29,30].

### 2.9. Statistical Analysis

Results were analysed using GraphPad 9.0 software (Prism, San Diego, CA, USA). The data are presented as means ± standard deviation (SD). To compare experimental groups, Student’s t-test was used. For multiple-group comparisons, one-way analysis of variance (ANOVA) followed by Tukey multiple-comparison tests were used. A *p* < 0.05 was established to distinguish statistical significance.

## 3. Results

### 3.1. Urine Cotinine Levels after SS and Smoke-Free Air Exposure

Urine cotinine concentration was used only as confirmation of cigarette smoke inhalation. While the average of basal cotinine levels (T0) was 0.66 ± 0.64 ng/g, in agreement with the literature for non-smokers [31], after SS exposure, urine cotinine levels were significantly increased. Compared to T0, urine cotinine was significantly increased at 1 h (2.58 ± 0.5 vs. 0.66 ± 0.64 ng/g, *p* < 0.00001), and it remained significantly stable after 2 h and 24 h (2.71 ± 0.85 and 2.5 ± 1.01 vs. 0.66 ± 0.64 *p* < 0.00001 and *p* < 0.001 respectively). Furthermore, SS urine cotinine levels were significantly increased compared to smoke-free air exposure: it was significantly increased at 1 h after SS exposure (T1, 2.58 ± 0.5 ng/g vs. 0.5 ± 0.09 ng/g; *p* < 0.0001), and it remained stable after 2 h and 24 h (T3, 2.7 ± 0.85 ng/g vs. 0.7 ± 0.01 ng/g and T4, 2.5 ± 1.01 ng/g vs. 0.6 ± 0.2 ng/g; both with * *p* < 0.05) (Figure 1).

### 3.2. NGF and Its Receptors TrKA and p75^NTR^ in WBCs after SS or SFA Exposures

The flow cytometric analyses showed that SS exposure induced a significant increase in the percentage and in the MFI of WBCs. In particular, the percentage of p75^NTR+^ WBCs was significantly increased during SS exposure, after 0.5 h (T1) and 1 h (T2), when compared to the same time points following control SFA exposure (T1, 18.95 ± 3.58% vs. 11.25 ± 0.95%, *p* < 0.005 and T2, 22.51 ± 5.76% vs. 10.25 ± 1.25, *p* < 0.0005). No significant differences were found at the other time points before and after exposure (Figure 2A). Moreover, the T1 also revealed a significant increase in MFI expression of p75^NTR^, comparing smoke-exposed and unexposed subjects (T1, 11.63 ± 3.13% vs. 6.79 ± 0.29%; *p* < 0.05) (Figure 2B). Peculiarly, p75^NTR^ appears to be an early biomarker of SS exposure, since there were no significant differences at late time points, in the percentage and in the MFI of TrKA-positive (TrKA^+^) WBCs at any time point investigated (Appendix A), or in the levels of serum NGF after SS exposure (Appendix A). Linear regression analysis showed a positive correlation between p75^NTR+^ WBCs and cotinine levels two hours after SS exposure (T2, r = 0.72, R^2^ = 0.5254, *p* < 0.0005) (Figure 2C), while no significant correlations were detectable between cotinine levels and p75^NTR^ expression at the early and late time points (Appendix A).

### 3.3. Impact of SS on WBCs: NGF Receptors and Inflammatory Cytokines

The gene expression analysis of p75^NTR^ revealed a significant increase at 1 h (T2) and 2 h (T3) (1.76 ± 0.45 and 1.77 ± 0.39, respectively) compared to smoke-free air exposure (T2 = 1.02 ± 0.10 and T3 = 0.97 ± 0.17) (* *p* < 0.05). No significant differences were found in the TrKA gene expression following SS exposure (Figure 3). qPCR revealed consistent, although not significant, differences in inflammatory cytokines (IL6, TNFα, and TGFβ) expression levels (Appendix A).

## 4. Discussion

The present study investigates the effects of short-term exposure to SS on the release of NGF and the modulation of its receptors TrKA and p75^NTR^ in healthy non-smokers.

The exposure to ETS has been classified as a “Group 1” carcinogen by the International Agency for Research on Cancer. Several adverse health effects on adults and children, including respiratory outcomes, acute and chronic cardiovascular effects, and lung cancer [3,9,32], have been connected to prolonged ETS exposure. Globally, 1.2 million deaths were attributable to ETS exposure, of which 63,822 occurred among children younger than 10 years old [9,33]. Although cigarette smoking was banned from most public places worldwide, very often non-smoker adults or infants are exposed to ETS at home, in private workplaces, and in vehicles.

Even if major pulmonary diseases are generally the result of long-term exposure to smoking [34,35,36,37], several studies have demonstrated that recurrent, short-term exposures may lead to the onset of health problems through unfavourable changes in the immune, cardiovascular, and endocrine systems. Several studies have investigated the hazardous health effects of chronic ETS exposure [38,39], while studies on the effects of acute, short-term ETS exposures are fewer [13,40]. As an example, long-term exposure to ETS is associated with an increase in several circulating markers of inflammation in non-smoker children and adults [41,42]. In contrast, the acute effects of short-term ETS on the inflammatory system are much less well-defined.

In this context, we have previously shown that NGF and its receptors have an important involvement in active smoking exposure. In particular, we have clearly evidenced a correlation between cigarette smoke and p75^NTR^. In fact, we have shown that, in healthy subjects, a smoking habit positively correlates to the increase in p75^NTR^ in peripheral blood mononuclear cells [14]. Furthermore, we have shown that an in vitro long-term exposure of human bronchial cells (BEAS-2B cell line) to cigarette smoke extracts induces the release of NGF and increases p75^NTR^ expression in these cells [15]. NGF is upregulated in the inflamed tissues of many diseases, and it can also directly influence innate and adaptive immune responses. The expression of both receptors (TrKA and p75^NTR^) is dynamically regulated in immune cells, inducing a variety of effects that can be either proinflammatory or anti-inflammatory [43]. In fact, TrkA activation promotes anti-inflammatory pathways, while p75^NTR^ promotes proinflammatory mechanisms contributing to chronic tissue inflammation [43].

Starting from this evidence, in the present study, we analysed NGF and its receptors in a group of non-smokers, adult volunteers exposed to short-term SS in a controlled home-like environment [27,44]. Determination of cotinine levels in urine samples was used as the classical, standard biomarker of SS exposure, as described in the literature [31,45]. Although cotinine is commonly used as a biomarker to validate self-reported smoking status, the selection of optimal cotinine cut-off values to distinguish smokers and non-smokers and the fluctuation of this analyte reveals a lack of standardization among studies [46]. The positive correlation between p75^NTR+^ WBCs percentages and cotinine levels following 1 h of SS exposure marks this NGF receptor as a potential novel molecule to be validated as a biomarker in further studies. We propose the evaluation of p75^NTR+^ in peripheral blood cells as a novel method to determine the brief exposure to cigarette smoke (e.g., in a forensic context).

Interestingly, our data show that, even after a very short period, SS exposure leads to a significant increase in the p75^NTR^ expression levels in WBCs, without any changes in TrKA expression levels and serum NGF concentration. SS exposure doubles the percentage of p75^NTR+^ WBCs, even at the very early time points (0.5, 1, and 2 h after SS exposure), although p75^NTR^ MFI increases at 1 h after SS exposure. These results were also confirmed by the increase in p75^NTR^ gene expression at 1 and 2 h after SS exposure. Altogether, these data show that short-term exposure to SS stimulates circulating WBCs to increase the levels of the proinflammatory p75^NTR^ without any changes in expression of TrKA, which conversely dampens the inflammatory response and limits tissue damage [43]. These data describe novel aspects of the roles/effects of NGF and of its receptors and contributes to the knowledge of the potential mechanisms during the early phases of short-term exposure to cigarette smoke, especially at home where smoke-free legislation cannot protect both adult non-smokers and, mainly, children.

This study shows that one hour of SS exposure does not increase the intracellular levels of IL6 and TNFα, in contrast to Flouris et al., who have demonstrated a significant increase in the levels of several inflammatory cytokines after one hour [12,47]. This discrepancy could be easily explained by the fact that the authors used a bar/restaurants’ smoking environment, with CO levels of 23 ppm, while we used a home-like simulation protocol, with 14 ppm of CO. This comparison appears interesting since it may lead to the hypothesis that, in certain circumstances, if we only use IL6 and TNFα as inflammatory markers, we could erroneously conclude that there is no inflammation, while the proinflammatory p75^NTR^ is already increased. Future studies may investigate this hypothesis.

## 5. Conclusions

Short-term exposure to SS induces a very early modulation of p75^NTR^ expression in WBCs without any modification in NGF concentration in sera or in IL6 and TNFα. Our study confirms the relationship between cigarette smoke and the inflammatory p75^NTR^, contributing to the knowledge of the mechanisms and pathway involved in the very early stages of short-term exposure to cigarette smoke and their impact on the health of non-smokers.

## Figures and Tables

**Figure 1 ijerph-19-10317-f001:**
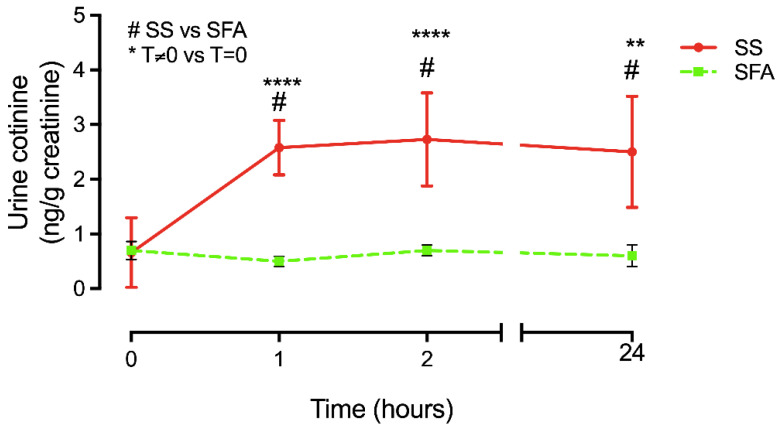
Urine cotinine levels after Side-stream Smoke (SS) or Smoke-Free Air (SFA) exposures. Urine samples from the subjects exposed to SS (red line and circles) or to SFA (green line and squares) were collected at different time points, as described in Materials and Methods sections. Urinary cotinine concentrations were measured by LC/MS and the final concentrations of cotinine for each time point were expressed in relation to creatinine levels, in order to normalize the results at the different dilution grades of the urine samples. The results are expressed as means ± SD. Statistical significances were determined using One-way ANOVA comparing each timepoint of the different experimental conditions (**** *p* < 0.0001, ** *p* < 0.001) and comparing each timepoint of the subject exposed to SS with their baseline (# *p* < 0.05).

**Figure 2 ijerph-19-10317-f002:**
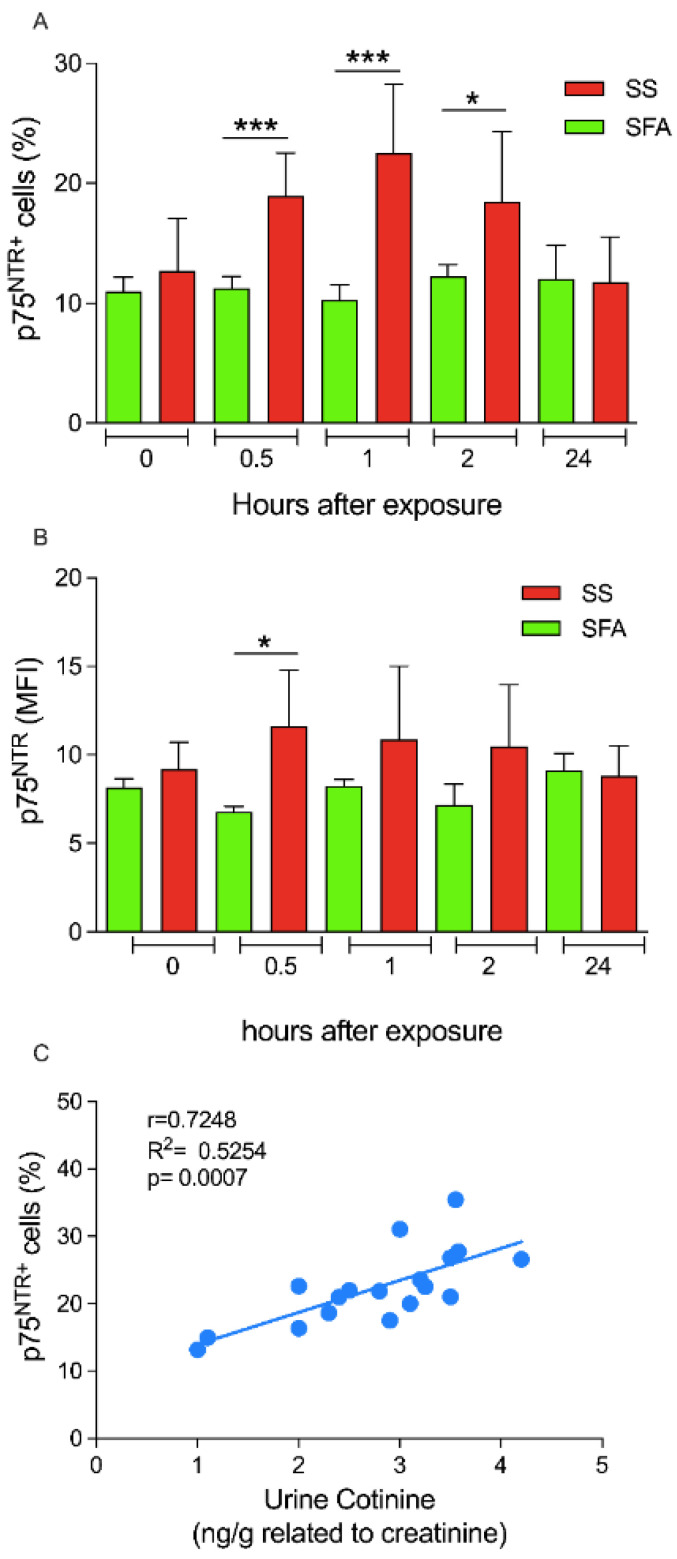
p75^NTR^^+^ WBCs distinguish SS- from SFA-exposed subjects. WBCs isolated from subjects SS-exposed or not were collected, analysed, and compared as described in Materials and Methods sections. The p75NTR+ WBCs are expressed as percentage of positive cells (**A**) and as MFI expression (**B**). Data are presented as mean ± SD. Statistical significances were determined using an unpaired *t*-test * *p* < 0.05, *** *p* < 0.0005 vs SFA-exposure. (**C**) Linear regression between the percentage of p75NTR+ WBCs and urine cotinine levels (ng/g) (r = Pearson index; R^2^ = coefficient of determination).

**Figure 3 ijerph-19-10317-f003:**
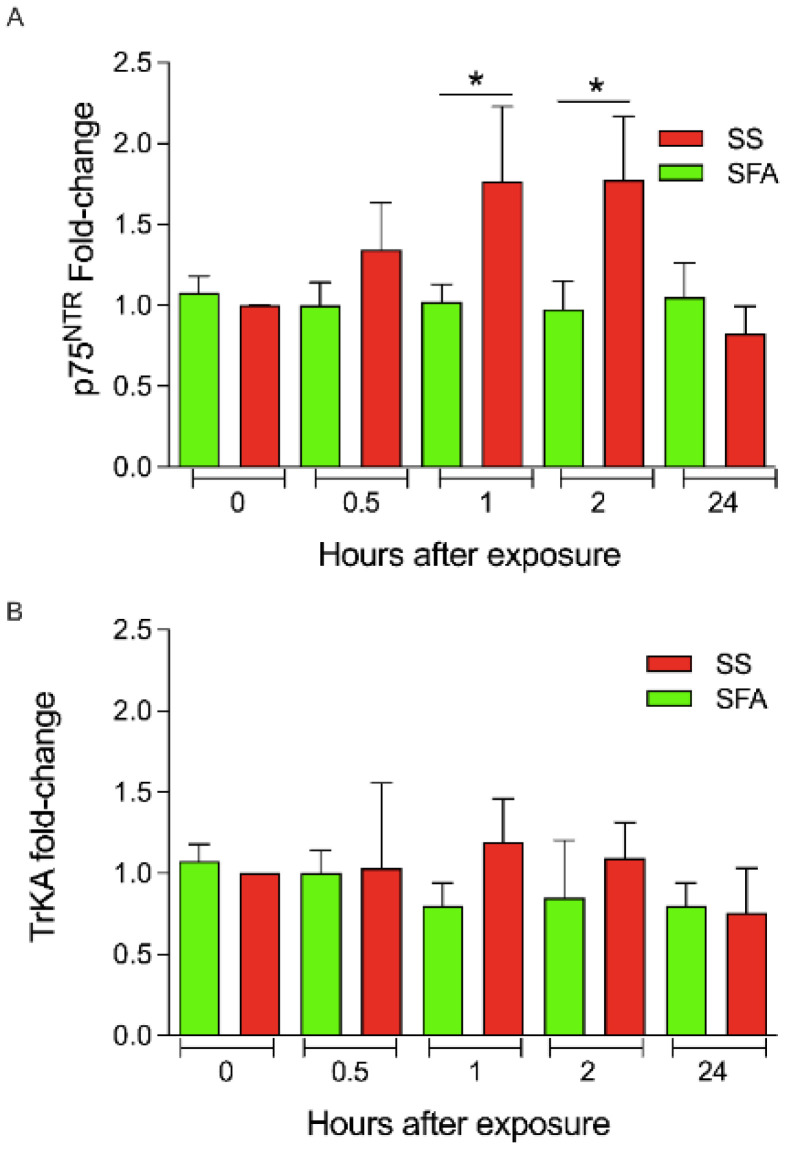
(**A**,**B**) Gene expression levels of NGF receptors. Gene expression levels of p75^NTR^ and TrKA were evaluated by qPCR using a specific primer set (Material and Methods section). GAPDH was used as endogenous control. Data are presented as mean ± SD. Statistical significances were determined using an unpaired *t*-test; * *p* < 0.05 vs. SFA exposure.

**Table 1 ijerph-19-10317-t001:** Demographic characteristics of the 21 participants in the study. Acronyms: SS = Side-stream Smoke exposed subjects; SFA = Smoke-Free Air exposed; BMI = Body mass index.

Characteristics	SS	SFA
Males/Females ratio, n/n (%/%)	9/8 (53/47)	2/2 (50/50)
Age (years), means ± SD	34.78 ± 11.21	31.50 ± 3.56
BMI (kg/m^2^), means ± SD	22.80 ± 3.17	22.68 ± 2.20

## Data Availability

Not applicable.

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
