# Peer review of "Short-Term Effects of Side-Stream Smoke on Nerve Growth Factor and Its Receptors TrKA and p75NTR in a Group of Non-Smokers"

_ijerph, 2022, doi:10.3390/ijerph191610317_

Round 1

Reviewer 1 Report

In this manuscript, the authors investigated the impact of short-term exposure to side stream (SS) smoke on NGF and its receptors TrKA and p75NTR. It was found that during the early phases, the SS exposure induced an increase both in the percentage of p75NTR+ WBCs, in their mean fluorescent intensity (MFI) and in gene expression. They also observed a positive correlation between the urine cotinine level and the percentage of p75NTR+ WBCs. These results add the knowledge of the mechanisms and pathway involved in the very early stages of a short-term exposure to cigarette smoke and their impact on health of non-smokers. From my personal opinion, it can be accepted for publication in IJERPH after addressing the following questions:

    1. Please don’t use abbreviations in the “Abstract” and “Keywords” parts.  2.The abbreviations should be defined upon the first mention. Please check all the abbreviations such as NGF, TrKA, p75NTR, etc.

 3.  Please check edits carefully. For example, In Figure2, 2 points of green line have no standard deviation (SD).

4.  Table1 is confusing. Why does it contain only 10 female participates, what the male participates’ data?

1.                

Author Response

Reviewer 1

From my personal opinion, it can be accepted for publication in IJERPH after addressing the following questions:

  1. Please don’t use abbreviations in the “Abstract” and “Keywords” parts.

2.The abbreviations should be defined upon the first mention. Please check all the abbreviations such as NGF, TrKA, p75NTR, etc.

We thank the reviewer; as suggested, we emendated all the abbreviations in the title and in abstract and keywords sections. We checked the rest of the manuscript defining the abbreviations where needed (introduction section).

  1. Please check edits carefully. For example, In Figure2, 2 points of green line have no standard deviation (SD).

We carefully checked finding that in Figure 1 two points of green line did not show standard deviation. Their SD were respectively 0.09 and 0.1 and for this reason overlapped with the symbols. We reduced the size of all the symbols and changed the colors of SD to address this issue.

  1. Table1 is confusing. Why does it contain only 10 female participates, what the male participates’ data?

As suggested, we modified table 1, showing males/females ratio instead of females number/percentage.

Reviewer 2 Report

Initial comments

This work, despite the fact that the “ Environmental tobacco smoke “ topic has already been extensively researched, adds many important and complex data that deserve to be published.

 Title

Short-term effects of side-stream smoke on NGF and its receptors TrKA and p75NTR in a group of non-smokers

Wouldn't it be better to spell it out    NGF ?

Nerve Growth Factor (NGF)

Abstract:

Comment:

Line 21……NGF

Please put in full…...Nerve Growth Factor (NGF)

1. Introduction  

 Comment:

It is suitable

2. Materials and Methods 

2.1. Study Participants 

Comment:

It is suitable

2.2. Home-Like Room Characteristics and SS Protocol

Comment:

It is suitable

2.3. Peripheral Blood collection and White Blood Cells isolation

Comment:

It is suitable

2.4. Serum Collection

Comment:

It is suitable

2.5. Liquid Chromatography–Mass Spectrometry (LC-MS) Analysis of Urine Cotinine

Comment:

It is suitable

2.6. Serum NGF quantification

Comment:

It is suitable

2.7. TrKA and p75NTR analysis by FACS

Comment:

It is suitable

2.8. TrKA and p75NTR and cytokines Gene Expression RT-PCR

Comment:

It is suitable

2.9. Statistical Analysis

Comment:

It is suitable

3. Results 

3.1. Urine cotinine levels after SS and smoke-free air exposure

Comment:

It is suitable

3.2. NGF and its receptors TrKA and p75NTR in WBCs after SS or SFA exposures

Comment:

It is suitable

3.3. Impact of SS on WBCs: NGF receptors and inflammatory cytokines

Comment:

It is suitable

4. Figures, Tables and Schemes

Comment:

It is suitable

5. Discussion

Comment:

It is suitable

6. Conclusions

Comment:

It is suitable

Supplementary Materials:

Comment:

It is suitable

References

Comment:

It is suitable

Thank you

Author Response

This work, despite the fact that the “Environmental tobacco smoke “ topic has already been extensively researched, adds many important and complex data that deserve to be published.

Title: Short-term effects of side-stream smoke on NGF and its receptors TrKA and p75NTR in a group of non-smokers. Wouldn't it be better to spell it out    NGF ?

Line 21……NGF; Please put in full…...Nerve Growth Factor (NGF)

We thank the reviewer; as suggested, we emendated he NGF abbreviation in the title. We also checked the manuscript defining the abbreviations where needed and where suggested by reviewer 1 (abstract, keywords and introduction sections).